# Whole exome sequencing study identifies candidate loss of function variants and locus heterogeneity in familial cholesteatoma

Ryan Cardenas[1], Peter Prinsley[2], Carl Philpott[1], Mahmood F. Bhutta[3,4], Emma Wilson[1], Daniel S. Brewer[1,5]☯*, Barbara A. Jennings[1]☯*

**1** Norwich Medical School, University of East Anglia, Norwich, United Kingdom, **2** ENT Department, James Paget University Hospitals NHS Foundation Trust, Great Yarmouth, Norfolk, United Kingdom, **3** Department of Clinical and Experimental Medicine, Brighton and Sussex Medical School, Brighton, United Kingdom, **4** ENT Department, Royal Sussex County Hospital, Brighton, United Kingdom, **5** Earlham Institute, Norwich Research Park, Norwich, United Kingdom

☯ These authors contributed equally to this work.
* b.jennings@uea.ac.uk (BAJ); d.brewer@uea.ac.uk (DSB)

## Abstract

Cholesteatoma is a rare progressive disease of the middle ear. Most cases are sporadic, but some patients report a positive family history. Identifying functionally important gene variants associated with this disease has the potential to uncover the molecular basis of cholesteatoma pathology with implications for disease prevention, surveillance, or management. We performed an observational WES study of 21 individuals treated for cholesteatoma who were recruited from ten multiply affected families. These family studies were complemented with gene-level mutational burden analysis. We also applied functional enrichment analyses to identify shared properties and pathways for candidate genes and their products. Filtered data collected from pairs and trios of participants within the ten families revealed 398 rare, loss of function (LOF) variants co-segregating with cholesteatoma in 389 genes. We identified six genes *DENND2C*, *DNAH7*, *NBEAL1*, *NEB*, *PRRC2C*, and *SHC2*, for which we found LOF variants in two or more families. The parallel gene-level analysis of mutation burden identified a significant mutation burden for the genes in the *DNAH* gene family, which encode products involved in ciliary structure. Functional enrichment analyses identified common pathways for the candidate genes which included GTPase regulator activity, calcium ion binding, and degradation of the extracellular matrix. The number of candidate genes identified and the locus heterogeneity that we describe within and between multiply affected families suggest that the genetic architecture for familial cholesteatoma is complex.

## Introduction

Cholesteatoma is a disease characterized by the proliferation of a pocket of keratinizing epithelium arising from the lateral tympanic membrane, and invading into the middle ear,

**Data Availability Statement:** All raw exome sequencing data and the combined VCFs for SNPs

and Indels detected using two pipelines (GATK HaplotypeCaller and Freebayes) are available at the European Genome-Phenome Archive (https://ega-archive.org/; Series accession ID: EGAS00001006147; Dataset accession ID: EGAD00001008671). The accession IDs for data from each sample are listed in Table 1 and are: EGAN00003527778, EGAN00003527779, EGAN00003527738, EGAN00003527740, EGAN00003527739, EGAN00003527754, EGAN00003527756, EGAN00003527737, EGAN00003527736, EGAN00003527741, EGAN00003527742, EGAN00003527743, EGAN00003527752, EGAN00003527751, EGAN00003527750, EGAN00003527762, EGAN00003527755, EGAN00003527753, EGAN00003527757, EGAN00003527759, EGAN00003527770, EGAN00003527774, EGAN00003527771, EGAN00003527773, EGAN00003527766, EGAN00003527747, EGAN00003527749, EGAN00003527748, EGAN00003527746, EGAN00003527745, EGAN00003527772, EGAN00003527744, EGAN00003527769, EGAN00003527768, EGAN00003527765, EGAN00003527767, EGAN00003527781, EGAN00003527780, EGAN00003527764, EGAN00003527763, EGAN00003527760, EGAN00003527761, EGAN00003527758, EGAN00003527775, EGAN00003527776, EGAN00003527777 The VCF IDs for each samples are listed in Table 1 and are: EGAZ00001862733, EGAZ00001862737, EGAZ00001862745, EGAZ00001862744, EGAZ00001862736, EGAZ00001862742, EGAZ00001862741, EGAZ00001862749, EGAZ00001862747, EGAZ00001862746, EGAZ00001862738, EGAZ00001862748, EGAZ00001862740, EGAZ00001862734, EGAZ00001862732, EGAZ00001862750, EGAZ00001862735, EGAZ00001862739, EGAZ00001862730, EGAZ00001862731, EGAZ00001862743 The metadata associated with each sample is available in Table 1. In combination this is the minimal data set needed to replicate all study findings reported in this article.

**Funding:** PP CP BJ Bernice Bibby Grant number A1136 no PP CP BJ Rosetrees Trust Grant number R203056 https://rosetreestrust.co.uk/project-grant-applications/ no PP Modi pump priming grant Royal College of Surgeons https://www.rcseng.ac.uk/dental-faculties/fds/research/fds-pump-priming-grants/ no The funders had no role in study design, data collection and analysis, decision to publish, or preparation of the manuscript.

leading to a progressive destructive lesion that erodes bone of the middle and inner ear [1]. Cholesteatoma can only be cured by microsurgical excision, and most patients suffer lifelong hearing loss due to the disease and/or the surgery. Although classified as a rare disease, there are over 7000 operations for cholesteatoma each year in the UK [2]; and a mean annual incidence of 9.2 per 100,000 was reported for surgically treated cholesteatoma in Finland [3] over ten years.

The aetiology of cholesteatoma is uncertain. Chronic otitis media in childhood is a predisposing factor, but only a small proportion of those with chronic otitis media will develop cholesteatoma [4, 5]. Animal models confirm the role of chronic mucosal inflammation in inducing cholesteatoma [6–8] but have also failed to illuminate how or why this occurs. Cholesteatoma grows as a self-perpetuating mass into the middle ear with activation of local osteoclasts, possibly as a result of an infection within the lesion [9]. The outer epithelial layer of the tympanic membrane has the unique property of centrifugal migration: carrying debris toward the outer ear canal [10]. Many theories have been presented about the pathophysiology of cholesteatoma and how it should be sub-classified; it has been called a pseudo-neoplasm but is perhaps more accurately described as an abnormal wound healing process [11]. In their review, Olszewska *et al.* [11], identified key clinical and histological features of cholesteatoma that warranted further research; these include disease recurrence, invasion, migration, hyperproliferation, altered differentiation, increased apoptosis, and the infiltration of stroma with immune cells.

Studies of differential gene expression of cholesteatoma compared with control tissue samples have been used to investigate underlying molecular and cellular pathology [12–17], through immunocytochemistry, PCR, microarray analysis, and RNA sequencing. Candidate-gene approaches (analysing molecules known to regulate pathways altered in cholesteatoma) have found increased expression of interleukin-1 (IL1), tumor necrosis factor-alpha (TNFα), and defects in the regulation of epidermal growth factor receptor (EGFR) [11]. Agnostic (hypothesis-free) transcript analyses [14–16] have found several hundred genes differentially regulated in cholesteatoma samples compared with normal skin, including pathways involved in growth, differentiation, signal transduction, cell communication, protein metabolism, and cytoskeleton formation, with a recent study identifying the proteins ERBB2, TFAP2A, and TP63 as major hubs of differential expression [16]. Studies of differential expression have been heterogeneous because of variations in tissue sampling and molecular detection. They also measure gene expression once cholesteatoma has formed, so may identify factors that result from the disease process rather than factors that initiate the disease. By contrast, genetic sequencing studies can identify constitutional or underlying risk factors, and therefore provide a route for studying causal biological pathways.

A clinical observation of familial clustering and the possibility of a heritable component for cholesteatoma was reported by one of the authors in 2009 [18]. A systematic review on the genetics of cholesteatoma identified 35 relevant studies, including case reports describing the segregation of cholesteatoma within families in a pattern consistent with a monogenic, oligogenic, or multifactorial trait [19], and in a recent survey, more than ten percent of cholesteatoma patients reported a positive family history [20]. Identifying functionally important gene variants associated with disease has the potential to uncover the molecular basis of cholesteatoma pathology, and whole exome sequencing (WES) can identify variants in coding DNA that co-segregate with the phenotype. We recently reported candidate loss of function (LOF) and missense variants in a pilot WES study of three affected individuals from a single family [21]. Here we build on this pilot to report findings from WES of ten additional families.

**Competing interests:** The authors have declared that no competing interests exist.

# Materials and methods

## Study design

This was an observational study to explore genetic associations for cholesteatoma within and between families. A linkage strategy was used to detect co-segregating variants in the exomes of affected individuals within each kindred. For WES, we selected the most distantly related participants within each family for whom we had extracted DNA, to reduce shared non-pathogenic variation filtering for bioinformatics analysis. In addition, we used an overlapping strategy to identify candidate genes of interest; that is, we identified genes with rare, loss-of-function (LOF) variants in two or more families. Further bioinformatic analyses were carried out to annotate candidate genes and variants of interest.

Our study objectives were

1. To establish a database of multiply affected families; to record their family histories (for otology and genetics); and to collect biological samples from participants for DNA extraction and storage in a biobank.

2. To undertake WES of selected affected individuals in the recruited families.

3. To deposit sequencing data and variant candidate filtering files (VCFs) in the European Genome-phenome archive (EGA).

4. To complete bioinformatic steps to filter for rare, functionally important variants within and between families.

5. To perform gene-level mutational burden analysis to identify genes that have a statistically higher proportion of deleterious mutations than would be expected in the general population.

## Setting, research governance, and participants

The study was approved by the East of England Cambridge Research Ethics Committee (reference REC 16/EE/01311, IRAS ID:186786), sponsored by the University of East Anglia, and registered on the National Institute for Health Research portfolio (CPMS ID 31548). Informed written consent was obtained from all participants. Participants were recruited from patients attending four hospital sites.

Inclusion criteria:

• Patients with a clinical diagnosis of cholesteatoma affecting at least one ear, and who have a family history of cholesteatoma.

• Families of patients in which there are one or more other affected individuals.

Exclusion criteria

• Only one affected individual with a confirmed case of cholesteatoma in the family.

• Families unwilling to consent to study participation.

A family history was collected from the index case of 10 families and any relatives who subsequently joined the study. For each family member recruited, we recorded on a REDCap [22] database the following: relationship to index case; date of birth; age at diagnosis and/or age at the time of surgery; unilateral or bilateral disease; secondary otology phenotypes; and diagnosis of genetic disease/congenital disorders.

## Biological samples and DNA extraction

Blood samples from 21 participants were collected in 3ml EDTA tubes and DNA was extracted using the QIAamp DNA Blood Mini Kit (Qiagen, UK). Samples were then quantified and checked for purity using a NanoDrop spectrophotometer (Thermo Scientific). All biological samples (blood and/or DNA) were stored by the Department of Molecular Genetics at the Norfolk and Norwich University Hospital. Before DNA extraction and quantitation were completed, samples were stored at 4 ˚C. Purified DNA was stored at—80 ˚C.

## Whole Exome Sequencing (WES): Library preparation, target capture, and sequencing methods

Two different service providers completed the next-generation WES and library construction from >500 ng of each high molecular weight DNA sample: the Genomics Pipelines Group at the Earlham Institute and Novogene (Cambridge, UK).

At the Earlham Institute, samples were processed using the NimbleGen SeqCap EZ Exome Kit v3.0 (bait library: SeqCap_EZ_Exome_v3_hg38) using an amended v5.1 protocol (NimbleGen 2015) producing 75bp paired-end reads and then sequenced on the Illumina HiSeq4000 platform. Libraries prepared by Novogene were processed using the SureSelect Human All Exon kit (bait library: S07604514 SureSelect v6) producing 180-280bp paired-end reads and sequenced on the Illumina NovaSeq 6000. Alignment statistics are described in S1 Table.

## Bioinformatics

**Alignment and variant calling.**   All tool versions and associated data files are listed in S2 and S3 Tables, respectively. Briefly, reads were mapped to the Human reference genome (GRCh38) using the sanger cgpMAP pipeline which utilises BWA-MEM [23]. All sequence data are stored in the European Genome-Phenome Archive (EGAD00001008671; EGAS00001006147; Table 1). Following quality control, SNPs and Indels were detected using two pipelines: one utilising GATK HaplotypeCaller [24] and the other FreeBayes [25] (S1 File). Variants were overlapped from both variant callers to give consensus on high-confidence variants for analysis.

**Variant filtering.**   Following alignment, variants were filtered using specific thresholds for several annotations, defined as hard filtering, for GATK and FreeBayes variant files (filtering parameters are detailed in S1 File). Variants were annotated for allele frequency using Slivar [26] which utilizes the Genome Aggregation Database (gnomAD) popMax AF [27] and the Trans-Omics for Precision Medicine Program (TOPMed) databases [28]. Variants were also annotated using the Ensembl variant effect predictor (VEP) tool giving SIFT/PolyPhen prediction for missense deleteriousness and PhastCons (7-way) for conservation scores. Variants with a population allele frequency ≥0.01 (1% in either gnomAD and TOPMed), a conservation score (PhastCons 7-way > 0.1), and predicted to be of functionally 'low impact' by Slivar [26] (https://github.com/brentp/slivar/wiki/impactful) were removed. Missense variants were annotated using SIFT [29] and PolyPhen [30]; those labelled to be 'benign' or 'tolerated' were excluded.

**Statistical analyses.**   In the family-based analyses, common variants shared between participants within a family were determined by intersecting the detected SNPs and Indels. Bcftools isec was used to identify identical SNPs. Indels were identified as identical if they overlapped by more than 10% using bedtools [31]. Families with greater than two samples were sequentially intersected to give indels with >10% across all family members.

**Table 1. Study participants.** Participants within families share numeric IDs. Age of diagnosis is given unless unavailable, where age at first surgery* is given instead. Cholesteatoma in both ears is described as bilateral disease (Y = yes) while disease in one ear is described as not bilateral disease (N = no). Familial relationships are described with respect to the index case. Sequencing data and VCFs were uploaded for each participant to the EGA data repository (EGAD00001008671; EGAS00001006147).

| Family ID | Subject ID | Age at diagnosis | Bilateral Disease | Sex | Index case or relationship to the index | EGA Accession | VCF accession |
|---|---|---|---|---|---|---|---|
| 1 | 1a | 28 | Y | Female | Sister | EGAN00003527778, EGAN00003527779 | EGAZ00001862733 |
| 1 | 1b | 30* | N | Male | Child | EGAN00003527738, EGAN00003527740, EGAN00003527739 | EGAZ00001862737 |
| 2 | 2a | 23 | Y | Male | Index | EGAN00003527754 | EGAZ00001862745 |
| 2 | 2b | 11 | N | Male | Brother | EGAN00003527756 | EGAZ00001862744 |
| 3 | 3a | 44* | N | Female | Index | EGAN00003527737, EGAN00003527736 | EGAZ00001862736 |
| 3 | 3b | 3 | N | Female | Child | EGAN00003527741, EGAN00003527742, EGAN00003527743 | EGAZ00001862742 |
| 3 | 3c | 6 | Y | Female | Sister | EGAN00003527752, EGAN00003527751, EGAN00003527750 | EGAZ00001862741 |
| 4 | 4a | 35 | N | Male | Index | EGAN00003527762, EGAN00003527755 | EGAZ00001862749 |
| 4 | 4b | 40* | N | Male | Brother | EGAN00003527753, EGAN00003527757, EGAN00003527759 | EGAZ00001862747 |
| 5 | 5a | 1 | Y | Female | Index | EGAN00003527770, EGAN00003527774, EGAN00003527771 | EGAZ00001862746 |
| 5 | 5b | 36 | N | Male | Child | EGAN00003527773, EGAN00003527766 | EGAZ00001862738 |
| 6 | 6a | 10 | N | Female | Index | EGAN00003527747, EGAN00003527749, EGAN00003527748 | EGAZ00001862748 |
| 6 | 6b | 5 | N | Female | Maternal aunt | EGAN00003527746, EGAN00003527745 | EGAZ00001862740 |
| 7 | 7a | 1 | N | Female | Index | EGAN00003527772, EGAN00003527744 | EGAZ00001862734 |
| 7 | 7b | 63 | N | Male | Maternal grandfather | EGAN00003527769, EGAN00003527768 | EGAZ00001862732 |
| 8 | 8a | 11 | N | Female | Index | EGAN00003527765, EGAN00003527767 | EGAZ00001862750 |
| 8 | 8b | 6 | N | Male | Brother | EGAN00003527781 | EGAZ00001862735 |
| 9 | 9a | 42* | N | Female | Index | EGAN00003527780 | EGAZ00001862739 |
| 9 | 9b | 44* | N | Female | Mother | EGAN00003527764, EGAN00003527763, EGAN00003527760 | EGAZ00001862730 |
| 10 | 10a | 1 | Y | Female | Index | EGAN00003527761, EGAN00003527758 | EGAZ00001862731 |
| 10 | 10b | 5 | Y | Female | Granddaughter | EGAN00003527775, EGAN00003527776, EGAN00003527777 | EGAZ00001862743 |

A gene-based mutation burden analysis was performed on individual samples utilizing TRAPD software [32], with the v2 gnomAD dataset providing a large and high-quality control cohort for analysis. Control positions with good sequencing depth (>10) in 90% of samples were used. Dominant and recessive models were determined by TRAPD software using the sample variant allele frequencies for cholesteatoma and gnomAD control samples. Two-sided Fisher's exact test was used to determine genes with enrichment in deleterious variants above the gnomAD background, as recommended by Guo *et al* 2016 [33].

Wilcox rank sum tests were performed using the rstatix (0.6.0) [34] package in R (version 3.1.4) [35]. Functional enrichment analysis was performed using gProfiler2 (v0.2.0) [36] utilising KEGG, Reactome, CORUM, and the GO Molecular Function database for terms. The gSCS (Set Counts and Sizes) correction method was used to determine significantly enriched pathways and ontology terms with significance $p < 0.05$.

# Results

## Participants

Twenty-one eligible participants were identified from our database who were members of ten multiply affected kindreds, Demographic, clinical features, and relationships between family members, are summarized in Table 1. Thirteen participants were female (13/21 = 62%) and six (6/21 = 29%) had bilateral disease at diagnosis or time of surgery. The median age for diagnosis or first surgical procedure for cholesteatoma was 11 (range 1 to 63). The participants within each kindred studied were either first-degree or second-degree relatives.

## Exome sequencing and the identification of variants

All DNA samples passed quality control steps, and Whole Exome Sequencing (WES) was completed for all 21 participants with an average of 75.1 million aligned reads per sample and a mean target coverage of 73.9X (S3 Table). Single nucleotide variants, insertions, and deletions were called using GATK and FreeBayes and filtered according to a hard filter. High confidence variants were produced by intersecting variants from both variant callers (Fig 1).

9,170,433 variants were detected using FreeBayes (8,048,428 SNPs; 316,886 Insertions; 440,166 deletions and 364,953 complex variants) and 631,501 using the GATK haplotype caller (598,794 SNPs; 14,490 Insertions; 18,106 deletions and 111 complex variants; Fig 1), with 229,645 variants detected by both approaches. Rare variants were retained based on a population allele frequency of less than 1% (gnomAD popMAX AF or TOPMed < 0.01) and a

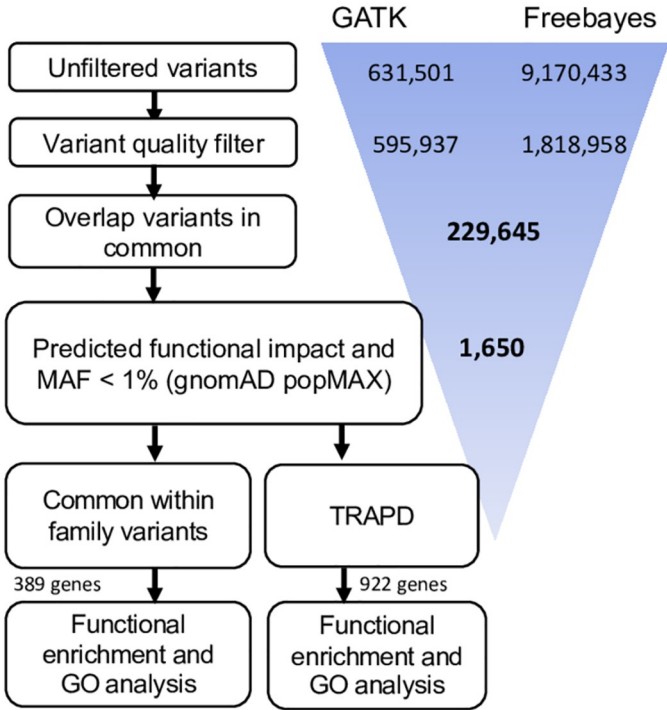

**Fig 1. Analysis overview.** Variants were called using GATK and FreeBayes, then filtered using a hard filter. High confidence variants were selected based on those that were detected by both variant callers. Variants were further filtered according to population allele frequency (retaining those < 1%) and predicted functional impact. Two distinct analyses were performed to identify potentially important genes, pathways, and ontology terms: 1) Identification of genes that have deleterious variants in multiple families; 2) A gene-based mutational burden analysis.

conservation score (PhastCons 7-way > 0.1). After further filtering for the most impactful and deleterious variants using Slivar's impactful filter (see methods), 1,650 variants remained (1,580 SNPs, 3 insertions, and 67 deletions).

## Variant filtering and family studies

Of the 229,645 variants initially detected, 30,294 variants are shared between affected individuals within families, which we identify as co-segregating shared variants (27,658 SNPs; 962 Insertions; 1661 deletions, and 13 complex variants). After filtering 398 high confidence, rare and deleterious variants occurring in 389 genes were identified (S1 Data). Of loci with co-segregating variants of interest, only six were found in more than one family (Table 2). Allele frequencies from gnomAD (median 0.002, IQR = 0.004), and TOPMed (median <0.001, IQR = 0.002), show these variants to be rare with the most frequent variant identified in only 0.5% of the general population. In addition, variants were shown to occur in highly conserved loci with 12/13 having a conservation score >0.9 (PhastCons7; Table 2).

Functional enrichment analysis revealed significant enrichment in 11 pathways or ontology terms (Fig 2; $p < 0.01$; Hypergeometric test; S2 Data) for the 389 genes where filtered co-segregating shared variants occurred. This included GTPase regulator activity (GO:MF), calcium ion binding (GO:MF), degradation to the ECM (Reactome), and USH2 complex (CORUM). Genes identified from functional enrichment analysis were only linked to a single family apart

**Table 2. A list of genes with co-segregating LOF variants in two or more families.** NCBI reference SNPs (rsID) give previously described variants. GnomAD (pop-MAX/ non-Finnish European—NFE) and TOPMed allele frequencies were used to give the proportion of variants in the general population: 1 indicates presence across all individuals in the general population and 0 a complete absence. SIFT and PolyPhen were used on missense variants to predict the impact on protein functionality. Phast-Cons-7-way conservation scores were determined for SNVs: 1 indicates complete conservation across 7 mammalian species and 0 as no conservation. The families for which a particular variant is present are listed in the final column by the family ID.

| Gene | rsID | GnomAD popmax AF | TOPMED AF | gnomAD NFE AF | Consequence | SIFT | PolyPhen | Conservation | HGVSc | HGVSp | Families |
|------|------|------|------|------|------|------|------|------|------|------|------|
| *DENND2C* | rs189506550 | <0.001 | <0.001 | <0.001 | missense | tolerated | probably damaging | 1 | c.842G>A | p. Arg281Gln | 1 |
| *DENND2C* | rs61753528 | 0.005 | 0.003 | 0.005 | missense | deleterious | probably damaging | 1 | c.2497T>C | p.Tyr833His | 10 |
| *DNAH7* | rs201273652 | 0.005 | <0.001 | <0.001 | missense | deleterious | probably damaging | 1 | c.3233A>T | p. Glu1078Val | 8 |
| *DNAH7* | rs115474479 | <0.001 | <0.001 | <0.001 | stop gained | NA | NA | 0.981 | c.6949C>T | p. Arg2317Ter | 2 |
| *NBEAL1* | rs199629983 | 0.004 | 0.001 | 0.001 | missense | deleterious | possibly damaging | 0.918 | c.5252G>A | p. Arg1751His | 9 |
| *NBEAL1* | rs180771101 | 0.003 | 0.002 | 0.003 | missense | deleterious | probably damaging | 1 | c.987T>G | p.Phe329Leu | 2 |
| *NEB* | rs201548700 | <0.001 | <0.001 | <0.001 | missense | deleterious | probably damaging | 0.999 | c.22187A>G | p. Lys7396Arg | 4 |
| *NEB* | rs114089598 | 0.005 | 0.003 | 0.004 | missense | tolerated | probably damaging | 0.999 | c.4649A>G | p. Lys1550Arg | 8 |
| *NEB* | rs764064217 | <0.001 | <0.001 | <0.001 | missense | tolerated | possibly damaging | 0.998 | c.6011T>C | p. Val2004Ala | 9 |
| *PRRC2C* | rs148813704 | 0.004 | 0.003 | 0.004 | missense | deleterious | benign | 0.986 | c.5980A>G | p. Asn1994Asp | 3 |
| *PRRC2C* | rs138220849 | 0.002 | 0.001 | <0.001 | missense | deleterious | benign | 1 | c.2191A>G | p.Met731Val | 2 |
| *SHC2* | rs201010410 | <0.001 | <0.001 | <0.001 | missense | deleterious | probably damaging | 0.991 | c.1595T>G | p. Leu532Arg | 3 |
| *SHC2* | rs768095487 | <0.001 | <0.001 | <0.001 | missense | deleterious | probably damaging | 0.274 | c.1510G>T | p.Asp504Tyr | 4 |

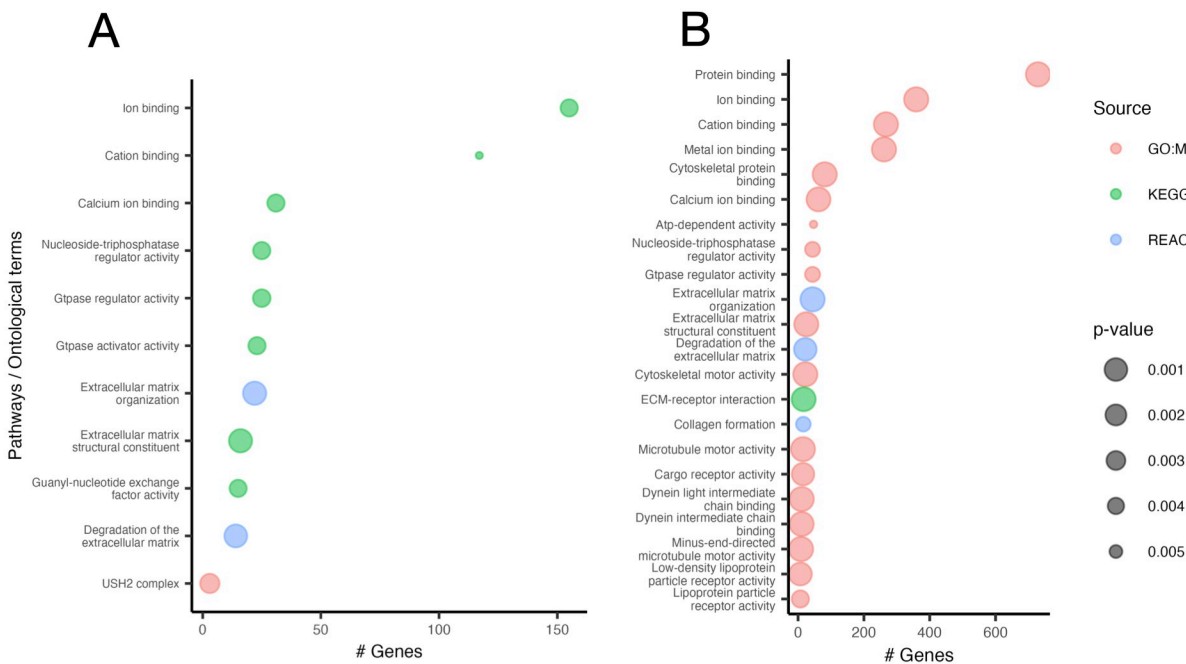

**Fig 2. Gene ontology and pathway analysis.** Performed on genes from filtered variants detected by the family overlap analysis in at least one family (A) and the TRAPD mutational burden analysis (B). Colours indicate the database used; (red) CORUM: the comprehensive resource of mammalian protein complexes, (green) GO MF: gene ontology for molecular function, and (blue) REAC: Reactome: the comprehensive resource of mammalian protein complexes. Dot size inversely indicates p-value. Only those terms with a *p* < 0.01 are shown (hypergeometric test). See S2 Data.

from *DENND2C* and *DNAH7* (*DENND2C*—family 1 and 10; *DNAH7*—family 8 and 2; Table 2)–within GTPase activator activity and calcium ion binding, respectively.

## Mutational burden analysis

We performed mutational burden analysis on the 1,650 variants that passed our strict filtering protocol (including those that were unique to individual members of a family). In the dominant and recessive analysis, we identified 910 and 12 genes respectively to be significantly enriched for deleterious variants in the cholesteatoma cohort compared to the gnomAD control cohort (Fig 3; S3 Data). Functional enrichment analysis revealed significant enrichment of affected genes in 17 pathways or ontology terms (Fig 2B, S4 Data), of which six were found in common with our previous analysis (Fig 4). These six included extra-cellular matrix (ECM) organization, GTPase activity, and calcium ion binding; each containing a larger number of associated genes in the mutational burden analysis compared to the family overlap analysis (Fig 4).

## Discussion

### Key results

The primary aim of this study was to identify candidate genetic variants that co-segregate with cholesteatoma within and between families. Bioinformatic analysis was used to annotate the genes of interest, which may have a role in cholesteatoma pathology. Data filtering collected from pairs and trios of participants within the ten families studied revealed 398 rare and damaging/deleterious variants in 389 genes (S1 Data) of which thirteen variants in six genes are of

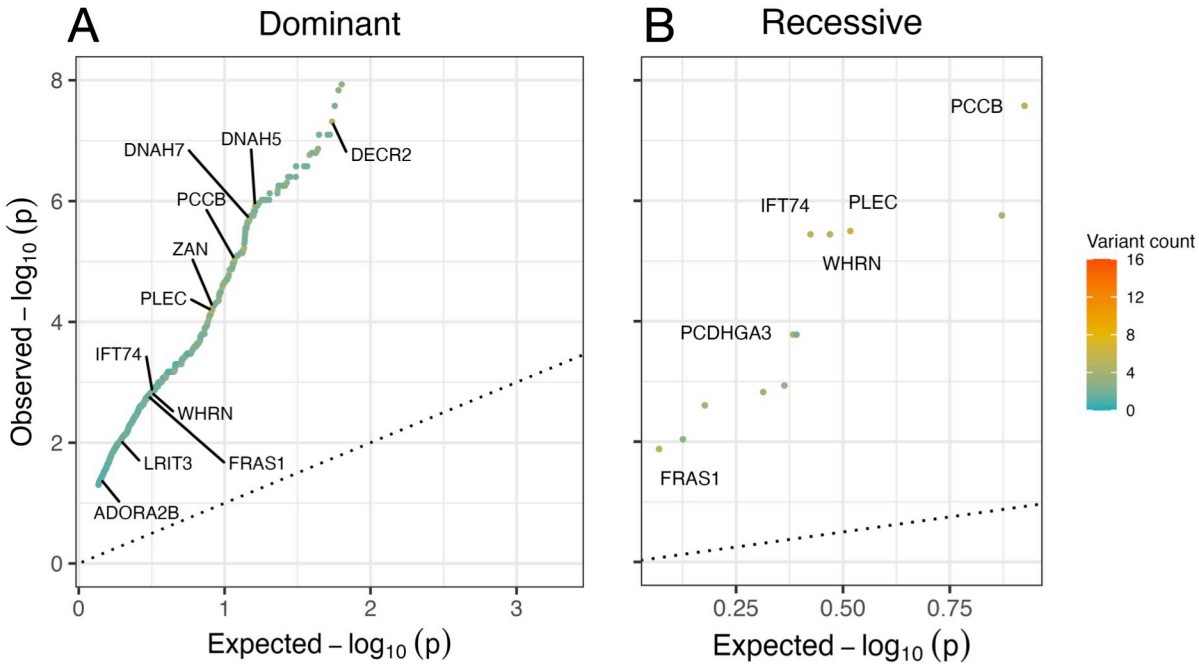

**Fig 3. Gene-based mutational burden analysis was performed on individual samples.** Based on allele frequencies from the cholesteatoma and control (gnomAD) cohort variants were split into dominant (A) and recessive (B) groups. The dot colour indicates the number of variants counted across the total cholesteatoma cohort, blue indicates a variant count of 0, and orange with a maximum count of 16. Statistical differences were determined using a two-sided exact Fisher's exact test ($p<0.05$). Points labelled with gene names have greater than 5 candidate variants in common across all samples. Refer to S3 Data for a comprehensive list of TRAPD genes.

greatest interest, because of overlap in two or three of the families (Table 2). These six genes: *DENND2C*, *DNAH7*, *NBEAL1*, *NEB*, *PRRC2C*, and *SHC2*, encode the following products respectively, DENN domain-containing protein 2C (a guanine nucleotide exchange factor); Dynein axonemal heavy chain 7 (a component of the inner dynein arm of ciliary axonemes); Neurobeachin-like protein 1 (thought to be involved in several cellular processes); Nebulin (a giant protein component of the cytoskeletal matrix); Protein PRRC2C (an intracellular protein required for stress granule formation); and SHC-transforming protein 2 (which is part of the ErbB signalling cascade).

The predicted impact of the listed variants on gene function, and genotype-phenotype correlations, can be used to infer their pathogenic potential. For example, in previous correspondence [21], we reported on the co-segregation of a stop-gained variant of the gene *EGFL8* (rs141826798) in a family with cholesteatoma, a gene previously associated with the common inflammatory skin disorder psoriasis, which has abnormal growth of the keratinizing epithelium in common with cholesteatoma.

No pathogenicity has been reported for the thirteen candidate variants identified from the overlap analysis (in their dbSNP database descriptions) [37]. One of the variants (rs115474479) is classified as an indel (stop gained) mutation in the gene *DNAH7*, the others are all classified as damaging/deleterious missense variants (Table 2). *DNAH7* variants are of interest because they encode a protein component of human cilia, where other functionally important mutations have been associated with primary ciliary dyskinesia (PCD). Cholesteatoma is associated with PCD [19, 38] and many children with PCD are treated for recurrent and chronic otitis media (COM) which in turn is an aetiological risk factor for cholesteatoma. Mutations in *DNAL1* and *DNAH5* are commonly reported in those affected by PCD, although

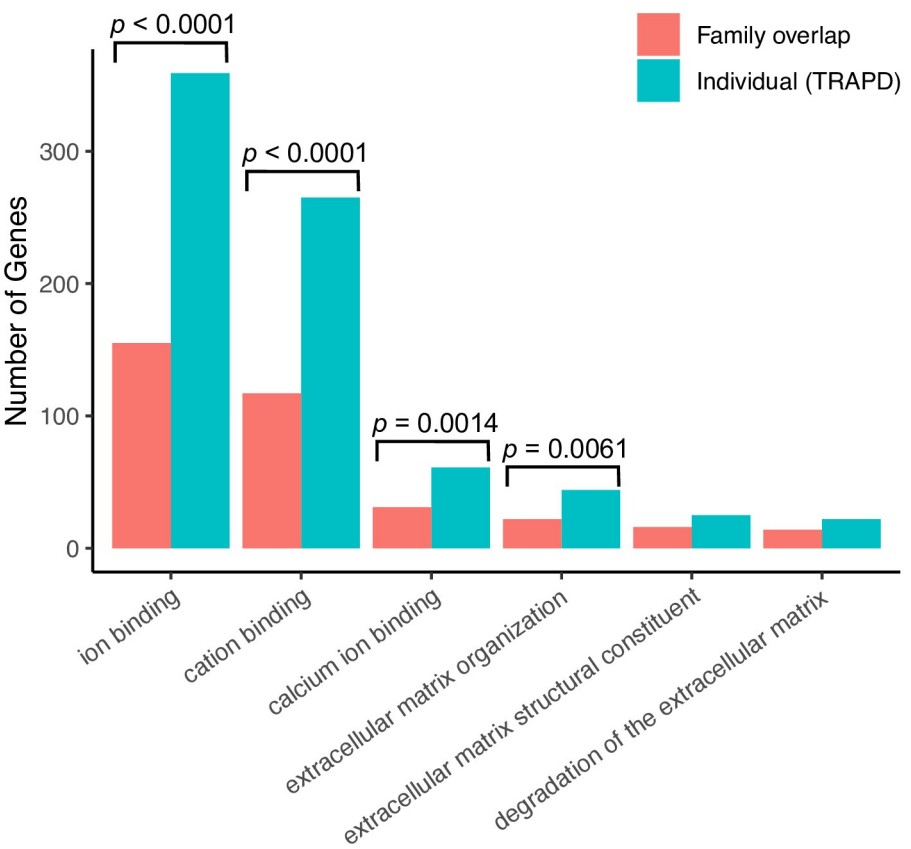

**Fig 4. Common pathways enriched.** Common pathway and ontology terms were found to be enriched for genes containing deleterious variants ($p < 0.01$; Hypergeometric test) in both the family overlap (red) and TRAPD (blue) analysis. The number of genes with deleterious variants in each pathway or ontology term is shown. Pathway and ontology terms where there is a significant increase in the genes associated with that pathway in the TRAPD analysis compared to the overlap analysis are highlighted ($p<0.05$; one-sided 2-sample test for equality of proportions with continuity correction).

some mutations in *DNAH7* (rs114621989 and rs770861172) have also been reported in PCD patients in the dbSNP database [37]. Damaging variants co-segregating in three families were identified in the very large gene, *NEB*, that encodes NEBULIN, an actin-binding cytoskeletal protein. *NEB* mutations typically cause inherited myopathies [39], but interestingly, cilia-related pathology could be associated with missense *NEB* variants because the process by which cilia form is dependent on the actin cytoskeleton [40]. These findings suggest that genetic factors that alter cilia structure and function may contribute to the development of some cases of cholesteatoma. Other non-constitutional risk factors and different disease pathways are inevitable given that most cases of cholesteatoma are sporadic cases and the complexity of the phenotype. A 2009 study of 86 individuals showed a reduced beat frequency of cilia in the middle ear of children with COM [41], but earlier smaller studies in such populations have shown conflicting results [42–44], and there is also debate whether any ciliary abnormalities found are the cause or effect of inflammation.

## A parallel analysis of mutation burden in the whole exomes

We supplemented our family studies with a gene-based mutational burden analysis to characterise genes with a higher proportion of mutations than observed in the gnomAD control

cohort [27]. This analysis focused on deleterious variants from individual samples over variants shared within families to take a more generalised approach, comparing the exomes from participants with cholesteatoma and control exomes. Fig 3 shows the results presented for a dominant model and a recessive model, highlighting the genes that were significantly enriched for loss of function (LOF) alleles in cholesteatoma individuals compared to the control. The significant mutation burden for the genes *DNAH5*, *DNAH7*, and *DNAH8* from the dynein axonemal heavy chain (DNAH) family provides further evidence for the relevance of ciliary abnormalities to the molecular pathology of cholesteatoma.

## Functional enrichment analysis

We also considered gene function through functional enrichment analysis to identify terms linked to candidate variants from the family overlap and mutation burden analyses. This analysis can highlight genes over-represented for biological processes, cellular localisations, and molecular pathways for gene products. Fig 2A illustrates the results of our functional profiling of gene lists carried out as part of the overlap analysis between families—common terms that were statistically enriched included GTPase regulator activity, calcium ion binding, and degradation of the ECM. ECM proteins, COCH and TNXB, were consistently down-regulated in cholesteatoma samples across several transcriptomic [14, 15, 45] and proteomic studies [46, 47]. In addition, several S100 genes known to regulate calcium binding and regulate ion channels show dysregulated expression patterns in cholesteatoma [14, 15, 45]. The agreement between cholesteatoma functional profiling and gene expression data suggests that the deleterious variants described are likely to have contributed to the disease.

## Interpretation and comparison with data from published transcriptomic studies

We compared our highlighted ontology and pathway terms from the family overlap study with terms identified from the studies described in our introduction [16, 17]. Significant and differentially expressed genes (DEGs) in cholesteatoma tissues were extracted from two previously published datasets to perform functional enrichment and GO term analysis. Imai *et al.* identified DEGs using RNA sequencing on a small cohort ($n = 6$) of cholesteatoma patients; a total of 733 genes were significantly downregulated. Jovanovic *et al.* analysed samples from COM patients ($n = 4$) and cholesteatoma patients with pre-existing COM ($n = 2$) which were analysed by microarray; 158 genes were significantly downregulated in cholesteatoma samples. In 8 of these genes identified as down-regulated in Imai *et al.* or Jovanovic *et al.* we detected a high confidence, rare and deleterious variant in our family-based analysis for at least one family. Similarly, in 12 genes we found variants in the mutational burden analyses. *CYP24A1*, *MUC16*, *MMP10*, *COL17A1*, *TJP3*, and *PPL* were identified in all three analyses (TRAPD, family overlap, and transcriptomics; S4 Table). Interestingly, *MMP10* and *COL17A1* are identified by the functional enrichment and GO analysis to regulate the degradation of the ECM, perhaps indicating the ECM has an important role in cholesteatoma aetiology. From a survey of cholesteatoma literature utilising transcriptomics, *MMP10* has been identified in 3 studies to be downregulated in cholesteatoma samples compared to the control tissues [15–17].

## Study strengths and limitations

We have achieved our objective to identify and share data about candidate genetic variants that co-segregate with cholesteatoma, and that may contribute to its pathology. We have provided a comprehensive and thoroughly annotated data set including links to our files in the EGA repository. The use of bioinformatic tools for mutation burden analysis and GO analysis

has provided additional evidence and curation about common biological processes, and identified molecular pathways and genetic variants associated with the risk of familial cholesteatoma that warrants further investigation. The rare deleterious mutations listed in S1 and S3 Data, from our family overlap and TRAPD analyses, are candidate variants of interest because they are predicted to be functionally important with respect to gene expression. As for most disease traits, we predicted that any genetic architecture (defined as the number and effect size of any contributing variants) would be complex for cholesteatoma. Heterogeneity in genetic risk factors is suggested by the number of co-segregating rare deleterious variants found in the family overlap and mutation burden analyses in this study and from our previous study [21]. We have identified a potential disease pathway for cholesteatoma development through the inheritance of genetic variants that alter cilia structure and function, and in pathways involved in cellular proliferation.

There are some limitations to discuss. We describe a hypothesis-generating observational study of exome data from 21 participants, so there is a risk of both false discovery (type 1 error) and missing variants of interest (type 2 error). Our primary study was small: it included only ten families and the filtering and quality assurance steps were stringent. Furthermore, our sample bank did not include DNA samples from many affected individuals from individually large pedigrees, limiting the reduction of shared non-pathogenic variation filtering for the individual family studies. We also only studied and curated exome sequences which preclude the identification of pathogenic variants in most non-coding regions of the genome. Our filtering and prioritization could result in pathogenic variants being discarded or overlooked. The rare minor allele frequency threshold of 1% was selected because cholesteatoma is classified as a rare disease; our approach would favour the identification of variants associated with a dominant inheritance pattern but could miss more common variants associated with a recessive model and or with complex genetic architecture. Therefore, our search for candidate pathogenic variants cannot be considered exhaustive and should be expanded in studies of large, affected pedigrees to identify more variants of interest, and to consider the penetrance of candidate variants. Our findings will now be applied to an analysis of sequencing data from a much larger cohort of individuals treated for cholesteatoma and recruited to the UK Biobank [48].

## Conclusions

Our WES studies of familial cholesteatoma cases identified candidate rare LOF variants in genes that encode products involved in ciliary structure, GTPase regulation, calcium ion binding, and degradation of the ECM. The locus heterogeneity suggests a complex genetic architecture for cholesteatoma, and we have identified molecular mechanisms and disease development pathways that warrant further characterisation.

## Supporting information

**S1 File. Supplementary methods.**
(PDF)

**S1 Table. Alignment statistics for DNA-seq exome samples.** The number of reads mapped to the hg38 assembly was calculated to give aligned and unaligned statistics. Exome target coverage was calculated using the manufacturer's bed files for DNA-seq library preps (see Material and methods). Maximum and mean coverage was calculated at target regions. The proportion of target regions with no coverage was also calculated.
(DOCX)

**S2 Table. Bioinformatics tools and versions used to process variants.**
(DOCX)

**S3 Table. A list of the files and their versions used by the bioinformatics tools.**
(DOCX)

**S4 Table. Underexpressed and mutated genes.** Genes identified from the family overlap and mutation burden analysis (TRAPD) were overlapped with genes that were significantly under-expressed in the transcriptomics studies from Imai *et al* (2019) or Jovanovic *et al* (2020).
(DOCX)

**S1 Data. Complete table for deleterious variants identified from the family overlap analysis.**
(CSV)

**S2 Data. Complete table for pathway and functional enrichment analysis for the family overlap analysis.**
(CSV)

**S3 Data. Comprehensive mutational gene-based analysis output.**
(CSV)

**S4 Data. Complete table for pathway and functional enrichment analysis for gene-based mutational burden analysis.**
(CSV)

## Acknowledgments

We are grateful to the family members who consented to participate in the study. Jane Woods, Olivia Whiteside, and our other partners in the UK National Institute of Health Research Clinical Research Network recruited study participants and collected samples and data included in this manuscript. Gavin Willis from the Department of Molecular Genetics, Norfolk and Norwich University Hospitals NHS Foundation Trust completed the nucleic acid extraction and quality assurance steps. Siham Mohamed contributed to the preliminary variant filtering steps during internal elective studies for her medical degree.

Some of the research presented in this paper was carried out on the High-Performance Computing Cluster supported by the Research and Specialist Computing Support service at the University of East Anglia.

## Author Contributions

**Conceptualization:** Peter Prinsley, Carl Philpott, Daniel S. Brewer, Barbara A. Jennings.

**Data curation:** Peter Prinsley, Carl Philpott, Barbara A. Jennings.

**Formal analysis:** Ryan Cardenas, Emma Wilson.

**Funding acquisition:** Peter Prinsley, Carl Philpott.

**Methodology:** Ryan Cardenas, Daniel S. Brewer, Barbara A. Jennings.

**Project administration:** Daniel S. Brewer, Barbara A. Jennings.

**Resources:** Peter Prinsley, Carl Philpott, Mahmood F. Bhutta, Barbara A. Jennings.

**Software:** Ryan Cardenas.

**Supervision:** Peter Prinsley, Carl Philpott, Daniel S. Brewer, Barbara A. Jennings.

**Validation:** Barbara A. Jennings.

**Visualization:** Ryan Cardenas, Emma Wilson.

**Writing – original draft:** Ryan Cardenas, Peter Prinsley, Carl Philpott, Daniel S. Brewer, Barbara A. Jennings.

**Writing – review & editing:** Ryan Cardenas, Peter Prinsley, Carl Philpott, Mahmood F. Bhutta, Emma Wilson, Daniel S. Brewer, Barbara A. Jennings.

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
