## [Decision Letter · Decision Letter 0]

6 Nov 2022

PONE-D-22-19738Whole exome sequencing study identifies candidate loss of function variants and locus heterogeneity in familial cholesteatomaPLOS ONE

Dear Dr. Cardenas,

Thank you for submitting your manuscript to PLOS ONE. After careful consideration, we feel that it has merit but does not fully meet PLOS ONE’s publication criteria as it currently stands. Therefore, we invite you to submit a revised version of the manuscript that addresses the points raised during the review process. Specific concerns:

Although the study is generally interesting, the reviewers raised a few concerns that must be clarified before the article is further considered for publication. Please make sure that all underlying data supporting the findings are made available and that all requirements of the journal are met.  Please submit your revised manuscript by Dec 21 2022 11:59PM. If you will need more time than this to complete your revisions, please reply to this message or contact the journal office at plosone@plos.org. Please include the following items when submitting your revised manuscript:A rebuttal letter that responds to each point raised by the academic editor and reviewer(s). You should upload this letter as a separate file labeled 'Response to Reviewers'.A marked-up copy of your manuscript that highlights changes made to the original version. You should upload this as a separate file labeled 'Revised Manuscript with Track Changes'.An unmarked version of your revised paper without tracked changes. You should upload this as a separate file labeled 'Manuscript'.

We look forward to receiving your revised manuscript.

Kind regards,

Rafael da Costa Monsanto, M.D.

Academic Editor

PLOS ONE

2.Thank you for stating the following financial disclosure:

“PP CP BJ Bernice Bibby Grant number A1136 no

PP CP BJ Rosetrees Trust Grant number R203056 https://rosetreestrust.co.uk/project-grant-applications/  no

PP Modi pump priming grant Royal College of Surgeons https://www.rcseng.ac.uk/dental-faculties/fds/research/fds-pump-priming-grants/ no”

“The study was funded by grants from the Rosetrees Foundation, The Royal College of Surgeons, and the Bernice Bibby Trust. We are grateful to family members who consented to participate in the study. Jane Woods, Olivia Whiteside, and our other partners in the UK National Institute of Health Research Clinical Research Network recruited study participants and collected samples and data included in this manuscript. Gavin Willis from the Department of Molecular Genetics, Norfolk and Norwich University Hospitals NHS Foundation Trust completed the nucleic acid extraction and quality assurance steps. Siham Mohamed contributed to the preliminary variant filtering steps during internal elective studies for her medical degree.

DSB acknowledges support received for the UEA Cancer Genetics team from Prostate Cancer Research, Prostate Cancer UK, Big C, and the Bob Champion Cancer Trust. Some of the research presented in this paper was carried out on the High Performance Computing Cluster supported by the Research and Specialist Computing Support service at the University of East Anglia.”

“PP CP BJ Bernice Bibby Grant number A1136 no

PP CP BJ Rosetrees Trust Grant number R203056 https://rosetreestrust.co.uk/project-grant-applications/  no

PP Modi pump priming grant Royal College of Surgeons https://www.rcseng.ac.uk/dental-faculties/fds/research/fds-pump-priming-grants/ no”

“NO”

Reviewers' comments:

Reviewer's Responses to Questions

**Comments to the Author**

1. Is the manuscript technically sound, and do the data support the conclusions?

Reviewer #1: Partly

Reviewer #2: Yes

2. Has the statistical analysis been performed appropriately and rigorously? 

Reviewer #1: Yes

Reviewer #2: Yes

3. Have the authors made all data underlying the findings in their manuscript fully available?

Reviewer #1: Yes

Reviewer #2: Yes

4. Is the manuscript presented in an intelligible fashion and written in standard English?

Reviewer #1: Yes

Reviewer #2: Yes

5. Review Comments to the Author

Reviewer #1: Recommend to be cautious when concluding anything from this very explorative study.

Recommend in line 403 and 403 to find an alternative and less strong word than 'Evidence'

The authors provide analyses within families, between families and between/across single participants. Which of these is the main priority of the study? Explain your motivation for- and discuss the implications of all three approaches (interpritation the results). Cholesteatoma mostly occurs sporadically - not inheritably. Everything that leads to chronic ear disease (e.g. mucosal or skin diseases) can indirectly increase the risk for cholesteatoma. This is a point that I recommend to add when discussing rare familial/inherited genetic variants with consequences for e.g. cilial function - and the link between this and cholesteatoma.

Emphasize, that this is an explorative study and discuss the false discovery rate/ likelyhood of incidental findings in these small groups.

Reviewer #2: The authors addressed familial cholesteatoma genetic architecture via whole exome sequencing (WES) for 21 individuals treated for cholesteatoma recruited from ten affected families. They searched for single nucleotide variants, insertions and deletions and performed a hard filtering to these variants to obtain high confidence variants. They fetched for variants that are shared between affected individuals within families. They performed gene-level mutational burden analysis and enrichment analyses. They detected the genes that showed rare loss of function (LOF) variants and identified six genes (DENND2C, DNAH7, NBEAL1, NEB, PRRC2C, and SHC2) which have LOF variants in two or more families. Functional enrichment analysis of the detected genes revealed 6 common pathways including calcium ion binding, extra-cellular matrix (ECM) organization, and GTPase activity. The missense variants were analyzed to predict the impact on protein functionality. All variants were classified as damaging/deleterious missense variants. One of DNAH7 variants (rs115474479) is classified as an indel (stop gained) mutation. Authors suggest that DNAH7 variants are of interest because they encode a protein component of human cilia, whose functional mutations have been associated with primary ciliary dyskinesia (PCD) which accompanies Cholesteatoma. The study used multiple bioinformatics tools in for analysis. There are some concerns for the improvement of the manuscript.

1. Please, mention the number of participants and number of corresponding families in the materials and methods section.

2. Would you please, mention the storage conditions of samples (line 160)?

3. Please, include fig S1 in the main figures as it shows the 6 common pathways of genes obtained by the two methods of analyses. Please, add a figure caption for it and unify the way of writing the labels (Upper case or lower case for the first letter).

4. Lines 321-323: ‘Calcium ion binding’ is repeated in the following part ‘’ These 6 included calcium ion binding, extra-cellular matrix (ECM) organization, GTPase activity and calcium ion binding, containing a larger number of genes for each term in the mutational burden analysis (Fig S1)'', so please correct it.

5. There are some typos and punctuation errors need to be corrected.

6. PLOS authors have the option to publish the peer review history of their article (what does this mean?). If published, this will include your full peer review and any attached files.

Reviewer #1: No

Reviewer #2: No

---

## [Author Response · Author response to Decision Letter 0]

20 Dec 2022

Please see the response to reviewers document.

---

## [Decision Letter · Decision Letter 1]

23 Jan 2023

PONE-D-22-19738R1Whole exome sequencing study identifies candidate loss of function variants and locus heterogeneity in familial cholesteatomaPLOS ONE

Dear Dr. Brewer,

Thank you for submitting your manuscript to PLOS ONE. After careful consideration, we feel that it has merit but does not fully meet PLOS ONE’s publication criteria as it currently stands. Therefore, we invite you to submit a revised version of the manuscript that addresses the points raised during the review process.

We look forward to receiving your revised manuscript.

Kind regards,

Rafael da Costa Monsanto, M.D.

Academic Editor

PLOS ONE

Journal Requirements:

Additional Editor Comments:

Please address the final comments made by the reviewers.

Reviewers' comments:

Reviewer's Responses to Questions

**Comments to the Author**

1. If the authors have adequately addressed your comments raised in a previous round of review and you feel that this manuscript is now acceptable for publication, you may indicate that here to bypass the “Comments to the Author” section, enter your conflict of interest statement in the “Confidential to Editor” section, and submit your "Accept" recommendation.

Reviewer #1: All comments have been addressed

Reviewer #2: All comments have been addressed

2. Is the manuscript technically sound, and do the data support the conclusions?

Reviewer #1: Yes

Reviewer #2: Yes

3. Has the statistical analysis been performed appropriately and rigorously? 

Reviewer #1: Yes

Reviewer #2: Yes

4. Have the authors made all data underlying the findings in their manuscript fully available?

Reviewer #1: Yes

Reviewer #2: Yes

5. Is the manuscript presented in an intelligible fashion and written in standard English?

Reviewer #1: Yes

Reviewer #2: Yes

6. Review Comments to the Author

Reviewer #1: No further comments.

Again, the comparison of findings with findings from other studies is very interesting. It would be interesting to know more about the roles of COCH and S100 proteins i cholesteatoma pathogenesis.

Reviewer #2: Comments

The authors have addressed all mentioned comments in a satisfying manner.

However, I have a couple of questions:

1) Regarding figure 4 “Fig 4. Common pathways enriched. Common pathway and ontology terms found to be enriched for genes containing deleterious variants (p < 0.01; Hypergeometric test) in both the family overlap (red) and TRAPD (blue) analysis. The number of genes with deleterious variants in each pathway or ontology term are shown.”

Is there a significant difference between the number of genes of family overlap (red) and TRAPD (blue) in each pathway? Can you show this on the bar chart and the error bars ?

2) According to your findings do you have any future perspectives concerning the study of genetic variants in cholesteatoma within and between families?

3) Please, revise the typos.

Example:

Our study objectives were

1. to establish a database of multiply affected families; to record their family histories

(for otology and genetics); and to collect biological samples from participants for

DNA extraction and storage in a biobank.

Put it in uppercase please.

7. PLOS authors have the option to publish the peer review history of their article (what does this mean?). If published, this will include your full peer review and any attached files.

Reviewer #1: No

Reviewer #2: No

---

## [Author Response · Author response to Decision Letter 1]

3 Feb 2023

We thank the reviewers for their comments. We have responded to each point raised by the reviewers below and detailed changes we have made to the manuscript.

Response to Reviewer #1

No further comments.

Author response: We thank the reviewer for taking the time to read through our manuscript and we are happy that our previous changes are acceptable.

Response to Reviewer #2

1) Regarding figure 4 “Fig 4. Common pathways enriched. Common pathway and ontology terms found to be enriched for genes containing deleterious variants (p < 0.01; Hypergeometric test) in both the family overlap (red) and TRAPD (blue) analysis. The number of genes with deleterious variants in each pathway or ontology term are shown.”

Is there a significant difference between the number of genes of family overlap (red) and TRAPD (blue) in each pathway? Can you show this on the bar chart and the error bars ?

Author Response: We thank the reviewer for their comment and have added additional information to Figure 4. Figure 4 uses gene counts for each gene ontology term or pathway. Each GO/pathway therefore only contains one count value (i.e. 358 genes from the TRAPD analysis in the ion binding pathway), therefore there is no spread to be visualised using error bars. We have performed an additional analysis comparing the numbers in the two analysis and adding onto Figure 4 the p-value for those pathways where there is a significant higher number of genes in the TRAPD analysis compared to the “family overlap”. We have also added the following detail in the figure legend:

“Pathway and ontology terms where there is a significant increase in the genes associated with that pathway in the TRAPD analysis compared to the overlap analysis are highlighted (p<0.05; one-sided 2-sample test for equality of proportions with continuity correction).”

2) According to your findings do you have any future perspectives concerning the study of genetic variants in cholesteatoma within and between families?

Author Response: We think the study of genetic variants in cholesteatoma within and between families should be expanded and complemented, given that any genetic architecture can be expected to be complex and heterogeneous.

In the final two sentences of our discussion, we express the need for more studies to identify candidate variants (given that our work to date has not been exhaustive for the reasons outlined in the limitations section). 

More work should include

• Family studies in large multiply affected pedigrees 

These could also identify more variants of interest through the use of WGS in addition to WES, and should be designed to consider the penetrance of candidate variants, through sequencing affected and unaffected individuals. These types of study would build on our own linkage approach that includes the most distantly related individuals and used WES rather than more comprehensive WGA.

• Cohort and case control analysis of unrelated, affected individuals to identify candidate variants of interest. 

As mentioned in the discussion, we are now using a GWAS approach with WES data collected by UK Biobank. 

Thank you for this question, we have added an additional clarifying sentence to the end of our discussion in response.

"Therefore, our search for candidate pathogenic variants cannot be considered exhaustive and should be expanded in studies of large, affected pedigrees to identify more variants of interest, and to consider the penetrance of candidate variants. Our findings will now be applied to an analysis of sequencing data from a much larger cohort of individuals treated for cholesteatoma and recruited to UK Biobank (51)."

3) Please, revise the typos.

Author response: We have corrected the typos identified and have found other typos by getting additional people to read the document and passing the document through two document checking algorithms (grammar.ly and LanguageTool). Hopefully, all issues have now been found.

---

## [Editor Report · Decision Letter 2]

9 Feb 2023

Whole exome sequencing study identifies candidate loss of function variants and locus heterogeneity in familial cholesteatoma

PONE-D-22-19738R2

Dear Dr. Brewer,

We’re pleased to inform you that your manuscript has been judged scientifically suitable for publication and will be formally accepted for publication once it meets all outstanding technical requirements.

Kind regards,

Rafael da Costa Monsanto, M.D.

Academic Editor

PLOS ONE
---

## [Editor Report · Acceptance letter]

14 Feb 2023

PONE-D-22-19738R2 

Whole exome sequencing study identifies candidate loss of function variants and locus heterogeneity in familial cholesteatoma 

Dear Dr. Brewer:

I'm pleased to inform you that your manuscript has been deemed suitable for publication in PLOS ONE. Congratulations! Your manuscript is now with our production department. 

Kind regards, 

on behalf of

Dr. Rafael da Costa Monsanto 

Academic Editor

PLOS ONE